# Developmental coordination disorder: "It's not on people's radars… They're not interested"

Rebecca Murray[ID][1]*, Cara E. Staniforth[1,2], Amira Shawak[1], Lucy H. Eddy[1,2,3]

**1** Department of Psychology, University of Bradford, Bradford, United Kingdom, **2** Centre for Applied Education Research, Bradford, United Kingdom, **3** School of Psychology, Northumbria University, Newcastle, United Kingdom

* r.murray2@bradford.ac.uk

## Abstract

### Background

Developmental Coordination Disorder (DCD) is an under-recognised neurodevelopmental disorder impacting 5–6% of children. There is plentiful research exploring the deleterious impacts of poor motor skill development, however there is a distinct lack of research gathering in-depth insights which explore the impact of DCD across life domains and the life course.

### Methods

Ten lived experience interviews (parents of a child with a diagnosis of DCD) were undertaken online, lasting 30–45 minutes. Participants were recruited both nationally and internationally to understand cultural differences. Parent interviews focused on experiences in primary care, education, friendships, and wellbeing. Data were transcribed and analysed using inductive thematic analysis.

### Results

Identified major themes highlighted (i) impact of DCD on the lives of children and their families; (ii) lack of knowledge and awareness surrounding DCD; (iii) inadequate support; and (iv) the role of diagnosis. All of this culminated in parents feeling the need to become advocates in the face of a profound sense of relative abandonment.

### Conclusion

Children with DCD and their families face multiple profound challenges in relation to accessing help and support for their difficulties. More needs to be done to facilitate synergistic support for children with DCD and their families across healthcare, education and the wider community. Effective cross-sector care will however, not be possible without the development of evidence-based training for all, which is underpinned by insider voices.

**Data availability statement:** Due to the sensitive nature of lived experience interviews, the research team believe that it would be inappropriate to make the data available. However, if anyone requests the data, we will provide heavily redacted transcripts - fully anonymised to ensure participant anonymity. Ethical

approval from the University of Bradford does not allow the data from this study to be shared without restriction. To access the data in an anonymised and redacted format, individuals can contact the University of Bradford Ethics Committee (ethics@bradford.ac.uk).

**Funding:** The work of the senior author (L.H. Eddy) and co-author (C.E. Staniforth) was supported by a grant from the Waterloo Foundation (ref: 27665413).

**Competing interests:** The authors have declared that no competing interests exist.

## Introduction

Difficulties integrating sensory information into skilled action are relatively common in children, with such challenges being classed as clinically significant in 5−6% of children [1]. These children often struggle with daily living skills such as dressing, personal hygiene, using cutlery, as well as engaging with physical activity and common education tasks such as handwriting [2,3]. Where you live in the world impacts on the diagnosis you would receive for such difficulties. For example, in the UK and parts of Europe this is diagnosed as Specific Developmental Disorder of Motor Function (SDDMF) or Developmental Motor Coordination Disorder (DMCD) via the World Health Organisation's International Classification of Diseases (ICD-10 [4]; and ICD-11 [5] respectively). In the United States of America (USA) and Australia, diagnosis tends to be made of Developmental Coordination Disorder (DCD) via the American Psychological Association's Diagnostic and Statistical Manual of Mental Disorders Fifth Edition (DSM-5; [6]). Adding to the complexity around labels, the term Dyspraxia used to be used and is still heavily used in education and community settings, particularly in the UK, despite this no longer being an accepted diagnostic terminology for early-onset sensorimotor difficulties [7]. International consensus was reached to use the term 'DCD' [8]; however, this is still inconsistently used. For the purpose of this paper, this group of diagnoses will be referred to as DCD hereafter.

This confusion around terminology has been highlighted as key barrier within diagnostic and support pathways for DCD [9,10], and is likely contributing to high levels of under-diagnosis [11]. Parents have also highlighted major dissatisfaction and stress associated with DCD pathways, highlighting long waiting times, inconsistent knowledge and substandard post-diagnostic support [10,12]. This is concerning, given that early intervention has been shown to be effective at improving outcomes for children with DCD [13]. Moreover, research has highlighted the long-term deleterious impacts of poor sensorimotor development on other aspects of development, which could be mitigated through early intervention. For example, research has highlighted the detrimental impact of poor sensorimotor skills on socioemotional wellbeing, including stress levels, psychological distress, symptoms of anxiety and depression, emotional reactivity and low self-esteem [14–16]. In addition, large scale epidemiological studies show children with poor sensorimotor development when they start school are less likely to perform well in formal academic tests [17,18]. Some research suggests this may be underpinned by cognitive differences [19].

Whilst studies looking at associations between poor sensorimotor development and other aspects of wider development are crucial, it is also important to consider the lived experience context of these impacts. There is a growing body of evidence consulting individuals with lived experience of DCD, including comprehensive questionnaire studies, like Impact for DCD which quantify challenges faced by those with DCD, highlighting a high proportion of individuals encountering (i) barriers along the pathway for assessment and support, (ii) difficulties engaging with physical activity and education and (iii) challenges with making new friends and associated socioemotional wellbeing [3,20]. There have been a few in-depth interviews and focus groups, however these tend to focus on singular aspects of living with DCD, such as crossing

roads [21]. There is therefore a need for more holistic in-depth interviews to gain insights from underheard voices. Given the role that parents play as advocates for their child, it is crucial to access the parental perspective within the realm of the health and wellbeing of their child(ren). This research aimed to reveal the reality of everyday life for children with DCD and their families via the voice of the parent – a reality which transcends symptomology to wider issues including awareness and support, from the viewpoint of parents.

## Methods

### Design

This research utilised a qualitative design, which allowed an exploration of lived experience (via interviews). This enabled an analysis, which presents a detailed holistic picture of the challenges faced by children with DCD and their families. Semi-structured interviews were used to enable a flexible approach which encouraged participants to share their experience, whilst also ensuring discussions were relevant for the research aims.

### Participants

Participants were recruited via purposive sampling, via adverts on social media (Facebook, Twitter, BlueSky, LinkedIn and Instagram) and through contacting all English-speaking national DCD/ Dyspraxia charities. Inclusion criteria required participants to have children with a formal diagnosis of DCD, and proficient levels of English to participate in interviews.

Ten parents of children with a diagnosis of DCD participated. Of these participants, two were from North America, one was from Asia and the remaining seven were based in the UK. Participants' children were diagnosed between the ages of 3 and 12 years. At the time of interview their children were between the ages of 8 and 29 years. Of the children discussed in the study eight were male and two were female. Of the parents included in this study seven had a child with co-morbidities, including three Autistic children, one child with ADHD and two that were both Autistic and had a diagnosis of ADHD, as well as one with Apraxia of speech. Given the high co-occurrence rates within academic literature, this was to be expected [11,22,23]. Participants were recruited from 6th January 2025–4th March 2025. Data saturation was achieved via the ten participants that were recruited, and therefore no further recruitment was required.

### Materials

Interview questions were generated to gain insights into stigma and challenges faced all aspects of life, including accessing diagnosis and aftercare within healthcare services as well as experiences related to family, parental support networks, child friendships and education. Therefore, the topic guide was intentionally broad to capture insights related to the challenges faced by children with DCD and their families according to previous literature [10,24], whilst also enabling a more holistic lens in order to explore underheard challenges. Interview questions can be found in Supplementary Material 1. These questions were not pilot tested. Prior to interviews participants were given the opportunity to access the interview guide.

### Procedure

Interested parents contacted the research team via email addresses provided on adverts. After sending through a participant information sheet, and answering any questions, participants that wanted to take part were invited to complete a written consent form, which was subsequently returned to the research team via email. Once consent had been obtained, a time and date were set for the online interview (via Microsoft Teams) at the convenience of the participants. All authors were involved in interviews. A minimum of two authors were present at all interviews to ensure all relevant questions were asked, and areas for discussion were fully covered. No one outside of the research team attended the interviews. Participants were asked seven questions, with prompts for further discussion used where

necessary. At the end of the set questions, participants were asked if there was anything else they would like to discuss in relation to their child's DCD diagnosis, to ensure their experiences were fully presented. Interviews lasted between 30 minutes to an hour.

All interviews were recorded and transcribed via Microsoft Teams. After the interviews, transcripts were downloaded, and the authorship team checked transcripts against recorded content, to ensure they fully reflected conversations. Participants and their children were given pseudonyms and any potentially identifiable information about participants was removed from transcripts to ensure anonymity. Once anonymised, transcripts were sent to participants for member checking to ensure they were reflective of the participants' true feelings, thus improving the credibility and trustworthiness of the data [25]. Participants had two weeks to review and amend transcripts, before the data were analysed. Ethical approval for this study was granted by the University of Bradford Ethics Committee (reference: E1026).

## Analysis

Within an interpretivist framework, thematic analysis [26] was used to analyse interviews following the six stages outlined by Braun and Clarke (2006) [26]: (i) Familiarising yourself with the data: Data from each interview were transcribed, and each transcript was read and re-read by all authors, which allowed initial ideas to be noted independently; (ii) Generating initial codes: Using NVivo, all authors systematically coded any interesting features within and across interviews; (iii) Searching for themes: Codes were collated into the themes – all relevant data was subsequently gathered to evidence the themes; (iv) Reviewing themes: Themes were reviewed to ensure codes were appropriate and there was sufficient data to demonstrate these more broadly; (v) Defining and naming themes: authors continued to refine the specifics of each theme; (vi) Producing the report. The quotes included in the results section were selected based on relevance to research aims, and their vibrant and compelling nature, which effectively articulated the lived experiences which transcended participants' data. All authors were present when each transcript was analysed, which allowed consensus to be reached when disagreements and contrasting interpretations were tabled.

## Research team and reflexivity

The lead female author (RM; PhD) is a Lecturer in Psychology with expertise in qualitative methodology and underheard populations. CES (MSc) is a Peer Research Associate – her expertise is in motor development. AS (BSc) was an undergraduate student at the time of data collection, working on this project for her dissertation. LE (PhD) was an Assistant Professor in Psychology at the time of data collection, with expertise in childhood motor development and DCD.

Prior to consenting, participants were sent an information sheet which detailed the reasons for conducting the research. At the start of the interviews, authors introduced themselves and explained their role in the project and their motivation for the topic. No formal rapport was built prior to the interviews, however participants were given contact details to liaise with the team to answer any questions they had beforehand.

## Results

The current analysis presents the treacherous road travelled by children with DCD and their families. The data transcends symptomology to wider system issues, including lack of awareness and effective support which impact the lived experience of health and wellbeing within the DCD diagnosis. Stories of abandonment, struggles to be heard, battles to access effective support across sectors, and the conflict of diagnosis in relation to stigma collectively revealed a society which is failing to scaffold vulnerable children and their families. A summary of themes and subthemes can be seen in Table 1.

The ***impact of DCD on the lives of children and their families*** was a primary theme within which challenges with sensorimotor skills were prominent, and profoundly impactful on their daily lives. *Activities of daily living* were commonly discussed, with parents noting their role within their child's support system.

**Table 1. Table of themes.**

| Major Theme | Subtheme(s) | Example data |
|---|---|---|
| Impact of DCD on the lives of children and their families | Activities of daily living | *We never said it out loud to him, but we were like, why is he not coping with cutting using a knife and fork? You know, wiping his bottom when he goes to the toilet properly or showering properly. I mean, he still now has to have help to shower. He'll be 12 in April* – Rachel |
| | Integrating sensory information | *Harry can't go out on his own because he doesn't understand the distance of cars coming* – Rachel |
| | Physical play and extracurricular activities | *The more tired he gets the more likely he is to stumble over his feet or do something ridiculous… [Although Leo is 8 years old] we do have a buggy, which [helps] if it's an extended walking school trip* – Amy |
| | Peripheral positions within the social context | *So he's very much aware of his limitations and actually because he's rubbish at kicking a ball and he can't catch balls and things like that the boys are less likely to involve him*– Amy |
| | Finding friendships within difference | *He always struggled to find friends and stay friends…He's brilliant with children younger than him. But it appears [making friends with] children his age is something he really struggles with because he says they judge him more. He found a friend who was a new child that came into school when he was year 4. He'd moved from India and that was brilliant because English wasn't his first language, so he struggled. Harry had struggles and they kind of clicked together, which actually was good because he said he didn't feel the only one left out. There was two of them struggling together* – Rachel |
| | Bullying | *We had a situation in primary school where my son would be in the playground, and one of his classmates would chase him down, would hold him wait until his brother and all his mates came and put a circle around them and then tackle him to the ground.* – Wendy |
| | Self-worth and confidence in relation to peers | *He never thought he was good enough. He knew there was something different, but he didn't know what it was. So, he's always been thinking "Oh well, I'm not good enough, so I can't be friends with that person. I'm never going to be good enough"* – Aminah |
| | Mental health and wellbeing | *And it got to the point that I actually only thought he'd had a breakdown. He was at rock bottom. We were having anxiety attacks which he'd never had. Panicking of crowds of going into school* – Rachel |
| | Unrecognised cognitive challenges | *The planning to get out the door before school? Well, of course he's forgetting something. You know, like that executive functioning piece. But then that that's also like how he [struggles to] plan a complex math problem* – Wendy |
| Lack of knowledge and awareness surrounding DCD | Teacher naivety | *The biggest challenges or blocks we've come up against is more individual teachers that have thought "it's a load of rubbish"* – Cindy |
| | Inadequate knowledge and awareness within Healthcare | *I have a family doctor. I don't think, knew about DCD either, but I made sure that when Jack was diagnosed that I gave him a whack of information.* – Wendy |
| | Community | *When DCD was mentioned, it was a totally new thing to me… but then as soon as I looked online, "Oh my God, this is Duncan"* – Cindy |
| | Parents emerged as advocates | *The teachers weren't supporting him and trying to get him involved, and it was only when I kept going into school saying "right we've got a problem now"* – Rachel |
| Inadequate support | Education system | *Even though the Occupational Therapists went in and did practice, you know, buttoning and lacing and things, it didn't translate to the dinner ladies not telling her off and not fastening her coat for her. And telling her she shouldn't need to have her coat fastened at the age of 11. And, you know, she was being lazy* – Rose |
| | Inadequate support within Healthcare | *We did get referred to an Occupational Therapist, who, quite honestly, were about as much use as a chocolate teapot… they saw us once and discharged us* – Amy |
| | Wider community | *I joined the Dyspraxia Foundation, but that's closed down now, hasn't it* – Cindy |
| The role of diagnosis | Self-acceptance and confidence | *When I said "do you remember when you struggle?" He goes, "yeah", I said. That's because of that [DCD]. And he was like, "oh, okay, so I'm not broken?" and I go "no, you're not broken"* – Rachel |
| | Burden of the label of DCD | *He didn't want anyone to know and I have to respect that… he said "I don't people to judge me" and I think that's his greatest fear.*– Aminah |

*We never said it out loud to him, but we were like, why is he not coping with cutting using a knife and fork? You know, wiping his bottom when he goes to the toilet properly or showering properly. I mean, he still now has to have help to shower. He'll be 12 in April* – Rachel

*All of that of just that coordination, something we take for granted like zipping his coat up. You know, putting on his tie. You won't believe in primary school in year six he had a proper tie, [we] thought he's not going to be able to do a tie. He did a tie… in year 6. Gets to high school and it's a Velcro thing around the back of his neck. He can't do the Velcro* – Cindy

*In Junior School I went in some of the time, when she was changing [for] for swimming, for example, I used to go with her and help her get changed so she could actually get into the swimming pool before the end of the lesson! And then as well coming out at the end so she didn't delay the school bus* – Rose

*I mean, the mess he makes at tea time. He used to have one piece of kitchen roll and now has two pieces of kitchen roll and a wet wipe… to clean himself, and then when he's done, it's all over [himself]. He has to change his top for tea time, because he wears a white shirt [for school]… obviously it's going to be all down him* – Cindy

In addition, parents noted challenges with *integrating sensory information* into skilful and timely movement, which can sometimes lead to dangerous situations.

*Harry can't go out on his own because he doesn't understand the distance of cars coming* – Rachel

*He doesn't really know where his body is in space and time* – Amy

*He's just a bit like a bull in a China shop. He isn't spatially aware… just kind of crashing to things* – Sarah

Parents also illuminated that their children had challenges with *physical* p*lay and extracurricular activities*, and the need for scaffolding in order to access such activities. This was discussed as inhibiting opportunities to engage in sport and physical activity.

*The more tired he gets the more likely he is to stumble over his feet or do something ridiculous… [Although Leo is 8 years old] we do have a buggy, which [helps] if it's an extended walking school trip* – Amy

*Because right from the early years, when he was at nursery, he couldn't balance on a beam. When I'd go in and observe him, what I'd find is all the other children running around and doing what they wanted to do and he couldn't, he needed the support he needed somebody hold his hand. Even when I used to take him to the park I had to sort of like, hold him and tell him what to do. So, but when I'd look at the other parents, they didn't have to tell their children, "Oh, put your foot here. Put your hand there this way, your bum needs to go here. Watch out for this. Watch out for that"* – Aminah

*He likes the look of football, but he dare not do it because he knows he can't kick a ball and he can't catch* – Amy

*She was really delayed developmentally, so on all the climbing frames we had to teach her where to put hands and feet and everything for a long time before she could actually climb* – Rose

Parents also highlighted that their child's challenges in relation to play and physical activity had a negative impact on peer perceptions and relationships, which resulted in exclusion from activities, and a *peripheral position within the social context*.

*So he's very much aware of his limitations and actually because he's rubbish at kicking a ball and he can't catch balls and things like that the boys are less likely to involve him* – Amy

*He really struggled with P.E. because he got to that point where the in P.E., when they were choosing groups, nobody would pick him because they knew he was not good at P.E.* - Aminah

*There would be observations about him being alone on the playground and it would be like "well, see, that's social avoidance, so it is autism". And it's like, no, actually this is a really natural representation of DCD because when you watch him, he literally starts in the playground with everybody else, but then he can't keep up or he wears out and he has to sit and rest and everybody else runs off.* - Wendy

The reality of embodying a peripheral position within the social context often meant that children with DCD were seen to gravitate towards other children with differences. Such connections provided greater opportunities to form meaningful *friendships within difference*.

*He always struggled to find friends and stay friends…He's brilliant with children younger than him. But it appears [making friends with] children his age is something he really struggles with because he says they judge him more. He found a friend who was a new child that came into school when he was year 4. He'd moved from India and that was brilliant because English wasn't his first language, so he struggled. Harry had struggles and they kind of clicked together, which actually was good because he said he didn't feel the only one left out. There was two of them struggling together* – Rachel

*He doesn't have a lot of friends, but he has two or three good ones. He has a friend who is vision impaired and a little boy who has cerebral palsy. So I think they kind of banded together because they were all excluded from the playground* – Wendy

*She had a few friends all the way through school, but they didn't always work out. She gravitated towards other children with learning disabilities and played with them* – Rose

Although children with DCD were able to find friendships within the realm of difference, the reality of their difference also resulted in negative experiences including *bullying*.

*We had a situation in primary school where my son would be in the playground, and one of his classmates would chase him down, would hold him wait until his brother and all his mates came and put a circle around them and then tackle him to the ground* – Wendy

*And then also with secondary school come phones …access to social media. So TikTok dances… if all the kids in the group are doing a dance and you have coordination problems and you can't follow the dance and everyone's sharing this video to, like, the whole grade, then everyone is replaying how uncoordinated you are, so it's not like a one time thing. It's actually caught on someone's [social media] and it's being circulated so this half year has been really difficult socially for him* – Yasmine

Peer perceptions and negative experiences with peer relationships were also discussed as having a detrimental impact on their sense of *self-worth and confidence in relation to their peers*.

*He never thought he was good enough. He knew there was something different, but he didn't know what it was. So, he's always been thinking "Oh well, I'm not good enough, so I can't be friends with that person. I'm never going to be good enough"* – Aminah

*So…it all kind of spirals into…how confident he feels about himself, relative to his friends … I don't think Billy necessarily wants… his friends to know, for example, that he goes to bed at, you know, seven/ eight o'clock every night* – Marvin

As a result of a culmination of the challenges discussed above, parents reported profound detrimental impacts on their child's *mental health and wellbeing*, which often presented as school-related anxiety.

*He'd started to feel "Something is wrong with me. What's wrong with me? Why am I at doctors and I'm not sick? You're all perfect. I'm not"… I'd pick up my kid from school and he'd be crying and saying he wants to just run away and not come back* – Wendy

*And it got to the point that I actually only thought he'd had a breakdown. He was at rock bottom. We were having anxiety attacks which he'd never had. Panicking of crowds of going into school* – Rachel

*I got the diagnosis for him was because he had a lot of anxiety at school and I knew there was something wrong- well, different about him* – Aminah

*He was literally falling apart every day* – Amy.

School-related anxiety may have been underpinned not only by their position within the social context, but also by the *unrecognised cognitive challenges* associated with DCD in relation to learning, memory and planning.

*And he was falling behind and behind, and it got to the point he was nearly three school years behind in his English, and two school years behind in his maths.*

*And when I found out I was like.. how is he going to cope going into secondary school [when he's] that far behind? He's just not going to cope* – Rachel

*Forgetting homework was a big issue because you know, she just hasn't got the memory. If she's in the middle of doing something, she can't remember that the homework is reading page 8, for example. If they just say read this page for homework and then carry on with the lesson, there's no way she'd remember that by the end of the lesson, and then she might forget to bring her books home anyway* – Rose

*The planning to get out the door before school? Well, of course he's forgetting something. You know, like that executive functioning piece. But then that it's also like how he [struggles to] plan a complex math problem* – Wendy

These challenges in relation to cognition were also found to translate into the home context, impacting on everyday activities, which could result in frustration for both children and their families.

*The main issues that he has with DCD in relation to how he interacts with the rest of the family is his executive functioning. So… he's just really forgetful, he has difficulty, you know, remembering to do stuff and that causes a lot of pain for the rest of the family, because you can explain it, but they still get frustrated… I guess the two balance each other out like he gets criticised and he gets sad, but then…he kind of forgets about it and moves on* – Yasmine

*We found a really good toothbrush that gives them a really good clean… but the planning of that is [is challenging]. He's over here. Then he's down here. Then he's up there. And I said, just spend 20 seconds there and 20 seconds there and 20 seconds there. The planning is not there* – Cindy

*He still finds school overwhelming at times and we have to give him a lot of support at home, especially with planning and organising, but he gets very tired* – Marvin

A second major theme identified was a ***lack of knowledge and awareness surrounding DCD*** which was central to the experience of parents. It was evident across all sectors that children's lived experience of DCD was impacted by (i) misconceptions; and (ii) ignorance. For example, *teacher naivety* was commonplace, with parents alluding to this resulting in a disregard of their child's needs.

*The biggest challenges or blocks we've come up against is more individual teachers that have thought "it's a load of rubbish"* – Cindy

*I went to school and they didn't really know what dyspraxia was. They were quite vague "oh everybody's different, every child is different"* – Aminah

*At the very first school… the first support teacher was quite resistant to the idea of DCD and [was] pushing quite strongly for other diagnoses that our medical team had ruled out, but [they] come with a big bucket of funding. And when I actually sat down with her and talked through DCD, she said, "Oh, my son is 22 and he quite possibly could have this"* – Wendy

*Just last week, the teacher just told him, look, I think that you're losing the ability to write so instead of writing these things, instead of writing your book reports on the laptop now, I'd like you to write it by hand and I'd like you to go back and rewrite all your other book reports as well… For a child who finds it difficult just to, you know, to even write a like half a page.... It was really difficult* – Yasmine

*Inadequate knowledge and awareness within healthcare* was also apparent, and posed a significant problem for families who experienced obstructions within the system designed to support their children.

*I have a family doctor. I don't think, knew about DCD either, but I made sure that when Jack was diagnosed that I gave him a whack of information* – Wendy

*From the private sphere it's not easy to find a therapist who cares about DCD. It's not really a thing… What I've consistently heard is look, DCD is a symptom, right? It's usually attached to something else, so what else is it that your son has?* - Yasmine

*The first paediatrician who told us that he wasn't meeting the criteria for autism, she was very much like, "Oh, DCD has to be comorbid with something." So we were spending a lot of time looking in this grey area. So she thought, well, I'll refer him (for autism) anyway. And I'm like, seriously, you've just told me this doesn't fit for him. And it's a four year wait list here to even get the assessment. So I was like, we need to focus on what he does have, not what he may be attached to* – Wendy

*When we go to places like the dentist, which is a big one, I don't even say the word dyspraxia. I say he's autistic* [even though he isn't] – Cindy

This lack of awareness transcended beyond healthcare and education, as it was evident amongst the *community*, including parents who alluded to doing extensive research after diagnosis to understand this unfamiliar condition.

*When DCD was mentioned, it was a totally new thing to me… but then as soon as I looked online, "Oh my God, this is Duncan"* – Cindy

*I had to Google what it was…because obviously I've heard of dyspraxia, but never DCD* – Amy

*Nobody in the family knew what dyspraxia was, so I was the fountain of knowledge because everybody was saying, "well what does it mean Cindy?"* – Cindy

*One of my nieces, has since been diagnosed with dyspraxia, but I don't think they [aunt and uncle] would have considered that if it hadn't been for Billy's diagnosis and the realisations of the sort of things we've talked about* – Marvin

*I just wanted a community because it's scary when you get told your son has a condition you don't know how to deal with it. I didn't know how to navigate the system. I didn't know you know how long is therapy? Is it forever like am I spending $2800 a week every week for the rest of his life? How do people do that?* - Yasmine

*I knew nothing at this point and all my friends' kids were just, kind of developing very typically, so I kind of felt on an island* – Sally

There was also an evident lack of understanding amongst the wider community, including family friends, which presented an additional battle for families who were frustrated by common misconceptions.

*I think you'd have to go into a lot of explanation for people to understand because I think the general perception still is clumsy* – Rose

*Harry is quite happy to tell people, but I don't think it makes a difference because they don't really understand…I don't think it's talked about enough, I don't think people understand what it is, to then understand what he struggles with, without you having to explain* – Rachel

*I mean, most of my friends, no one knows what it is. No one's even heard of it… A lot of people haven't even heard the term* – Sally

*Initially there's this attitude of, "oh, well, he just needs to play more sport."*

*It'*

*s so much more than motor and then it'*

*s all those secondary social and emotional pieces that a lot of people don'*

*t …really understand... So it's really understanding the complex picture and helping people understand that it's much, much more than a motor condition* - Wendy

*DCD is not a thing [here]. It's not a thing. It's not on people's radars and they're not interested* – Yasmine

Due to the profound lack of knowledge and awareness surrounding DCD, *parents emerged as advocates*. For example, parents not only had to generate or collate resources for schools and healthcare, but also actively educate others on both the reality of DCD for their child, and how to ensure an inclusive environment.

*Victoria, my wife, has done some training with the school, which made a massive difference…particularly with individual class teachers* – Marvin

*As a parent, I felt very alone. I didn't know anybody else with the condition. I was trying to cobble together resources and I found lots of great research out there. But there was really nothing written by a parent. As just one mum, it was very easy for them to put up barriers. But I can open the website and say "OK, I'm a parent advocate. I'm trying to help, not just my kid, but all these other ones."* – Wendy

*Oh my gosh, my child is having trouble walking and talking, and nothing really mattered to me anymore. So I ended up leaving my job and really focusing on her* – Sally

*[The family doctor said] "This might be happy happenstance, since I've just had a woman come in with a child with the DCD diagnosis and she's really overwhelmed. Can I give her your number?" I'm like, "oh my God. Give my number to everybody"* – Wendy

Parents also had to fight for the rights of their children when they were being discriminated against, which resulted in multiple visits to school.

*The teachers weren't supporting him and trying to get him involved, and it was only when I kept going into school saying "right we've got a problem now"* – Rachel

*I had to keep going into the school on a very regular basis to explain why Mary wasn't being lazy and she wasn't "not focusing" and why she wasn't, you know, why she was slow with things and couldn't do things. Every year I think I went in… [and] she was a very slow eater and she was constantly being chastised for being slow, and she was in the way when they came to clear the tables. So I had a bit of a battle all the way through school, just constantly having to go in and remind them…And I actually suggested putting really big gardening gloves on and trying to type because that's how it feels to Mary* – Rose

*So I have advocated for the removal of dodgeball, because the kids would just target him straight away and he's either out first go and he has to sit out the rest of the game, or he's just pummelled the whole time* – Wendy

*So I have advocated for the removal of dodgeball, because the kids would just target him straight away and he's either out first go and he has to sit out the rest of the game, or he's just pummelled the whole time* – Wendy

*Matthew will come home and say "my English teacher complained that my writing is really bad and I'm going to be assessed on it and he doesn't see me passing". And that is my cue to write to that specific teacher and say "actually he has this issue so can we not assess him on it?"* – Yasmine

Despite high levels of self-advocacy amongst parents, the reality of the support available for children was still substandard which revealed ***inadequate support*** as another major theme across (i) education; (ii) healthcare; and (iii) the wider community. For example, for many, their child's needs were neglected by those within the *education system* responsible for supporting their pupils.

*Even though the Occupational Therapists went in and did practice, you know, buttoning and lacing and things, it didn't translate to the dinner ladies not telling her off and not fastening her coat for her. And telling her she shouldn't need to have her coat fastened at the age of 11. And, you know, she was being lazy* – Rose

*I spoke to his teacher about my suspicions of dyspraxia when he was 6. The teacher had said she would discuss it with the SENDCo and get back to me. She never did* – Sarah

*They produced a laptop for her to use but some of the other children got jealous of her having a laptop... So the laptop was taken away…and she never saw it again* – Rose

*Some teachers have been hard. By the end of Grade 8 Physical Education (P.E.) [aged 13–14] we pulled him out and he now does P.E. independently online. Because the P.E. teacher in 2024 still picks kids to pick teams and he couldn't comprehend when my husband and I went in to meet with him, why that was not a good practise. His attitude was "well, the kids need choice". And I said, "My son can hear the other kids talking about leaving him to last and not picking him. How is that ok?"* – Wendy

Due to the lack of understanding and effective support within school contexts, many parents made the decision to either swap schools or pull their children out of mainstream education entirely. These decisions can come at great cost to families, if the decision is made that private education would be more suitable.

*We decided to put her into a small private school for secondary school [aged 11–16 years] because the thought of sending her to a bigger school, where she really didn't want to go, was going to be too difficult and she wouldn't get on very well at all. So we actually decided to pay to take her out of the education system* – Rose

*When I got his education health and care plan [support document which facilitates extra school funding], they said we can't meet his needs. They kept saying… we don't have the resources, we don't have the funding. That was the only thing that I kept being told, and they couldn't even do a reasonable adjustment plan, which is like the basic thing. So the best thing I ever did was move schools* – Aminah

*He's on his second primary school because his first primary school, [there was] no support across the board, basically, even though he desperately needed it* – Amy

*We made the big decision, because he didn't have an education health and care plan to take him out. And he's still out for now* – Rachel

*Inadequate support within healthcare* was also reported as being substandard, with limited resources available for parents after a diagnosis beyond leaflets and signposting. This lack of support resulted in a sense of abandonment amongst parents, who were expecting so much more.

*We did get referred to an Occupational Therapist, who, quite honestly, were about as much use as a chocolate teapot… they saw us once and discharged us* – Amy

*So they give you this diagnosis, but they don't tell you what you can do to improve it they don't… offer you extra appointments and teach you…what he needs to be doing every day* – Aminah

*A few leaflets, pamphlets to read through…pretty much stuff that you could just Google… rather than anything that would be child specific or you know about adaptations and things you might do in school* – Marvin

Parents also reported that healthcare professionals missed opportunities to identify their child's sensorimotor difficulties early on in their development, which led to a delayed diagnosis.

*I went to our paediatrician and they were very dismissive- they were like "no, this is fine, give it another couple months". And I was like, "I don't think so, but OK, maybe I'm overreacting". So then, you know, I had to circle back a few times… and I basically told the paediatrician "This is not going right, like she's not walking how she should be walking" and so she was kind of dismissive* – Sally

*I remember the health visitor coming…. she was asking questions like, "does he get up the stairs? Does he feed himself with a fork if he can't reach somewhere, will he pull a chair over and stand up on it?" And he did none of those things. And I said "no". And she said, "well, is he sort of making this movement?" And she wanted to tick the yes box* – Cindy

*[A Physiotherapist said] "It's fine. He's met the milestones. He's walking now. If you have any other concerns, just reach out to the public health service"* – Wendy

Parents revealed that there were fractured systems within public healthcare internationally, with long waiting lists, and increasing demands on services which meant care practitioners were not consistent across their child's journey. This lack of meaningful support within public healthcare also led to some parents paying for private diagnosis and intervention, often at a huge cost to the family.

*It's constantly passing [his care] from person to person I never really spoke to the same person twice* – Rachel

*My son was on the waiting list to see an occupational therapist for about five or six months…and when we got in there in front of an occupational therapist, she was amazed she was like "I can't believe you managed to get an appointment to see us…it's incredible, it's almost a miracle you're here"* – Yasmine

*The reason it took less time was because we paid for it privately and we were fortunate to be able to do that* – Wendy

*To do the private stream it was… $1800 for a 15-minute session with the play therapist, this was four years ago, and prices have gone up since then. And then it was another it was over $2000 to do the first initial meeting with the occupational therapist…And then it was nearly $20,000 to do the assessment. Yeah, it was prohibitively expensive, and it would have been out of the reach of a lot of families* – Yasmine

As parents alluded to ineffective support within healthcare and education, the *wider community* could play a pivotal role in supporting children with DCD and their families. However, discussions highlighted that historically there has been a distinct lack of resources available, something which continues to burden families today.

*I can't think of particular sources…I don't think there was a particular place or person or organisation…And it doesn't particularly feel like the resources that are dramatically improving… It's seven years since he was diagnosed, and it doesn't feel dramatically different in that regard, as to what's out there* – Marvin

*That's really hard to find [support]. I've not come across anything for dyspraxia…maybe on the website if you Google it… I've not come across any support groups* – Aminah

*I joined the Dyspraxia Foundation, but that's closed down now, hasn't it* – Cindy

The final theme which was **the role of diagnosis** illuminated both enabling and disabling aspects according to parents. Thus, although discussions painted a challenging picture for children with DCD and their families, parents revealed the potential for diagnosis to support the wellbeing of their children which facilitated *self-acceptance and confidence.*

*When I said "do you remember when you struggle?" He goes, "yeah", I said. That's because of that [DCD]. And he was like, "oh, okay, so I'm not broken?" and I go "no, you're not broken"* – Rachel

*When you have a diagnosis, I'm able to explain to him "you were always good enough. It's just that it would take you longer because of this…Normally, if it takes a child a month to do that, it might take you two months to do that and you would have would probably have to work a bit harder, but you're going to get there in the end" and that's all he really needed to hear* – Aminah

*With the diagnosis, as he's gone secondary school, being able to… take away some of the comparisons to other kids has been particularly good for his confidence* – Marvin

*And then I think one of the strongest, best outcomes of giving that language around DCD was we could then help himself advocate. So I've heard him say even some days "Oh my DCD is getting the better of me today." You know, if he gets his words muddled up or something's difficult. But then also in school settings he can say, "Look, hey, I've got DCD. I'm doing my best."* – Wendy

However, despite the positive impact of diagnosis on their sense of self, the burden of the label of DCD was also heavy for some children, who faced challenges in relation to stigma and a lack of acceptance from others both in and beyond the family.

*He didn't want anyone to know and I have to respect that… he said "I don't people to judge me" and I think that's his greatest fear* – Aminah

*It just it becomes very much, "He just needs to try harder." It becomes really frustrating for the kid with the invisible disability, the onus is on him to try harder* -Wendy

*So Matthew's father doesn't believe in the diagnosis…he doesn't think it's a thing* – Yasmine

## Discussion

This research aimed to reveal the reality of everyday life for children with DCD and their families, a reality which transcends symptomology to wider issues including awareness and support, from the viewpoint of parents. What was clear from the analysis was the wide-reaching impact that a DCD diagnosis had on both the child and their family. For example, the lack of independence in daily living was evident amongst families who often had to support their children in a way which was not age appropriate for self-care tasks, including showering and toileting. Although this is a well-established feature of the diagnostic criteria [4–6] and previous research has alluded to such issues [2,27], the personal stories within the current analysis revealed the hidden burden on (i) the child, who was cognisant that they are different to their peers; and (ii) the parent, who was central to their child's support system for basic needs.

The role of the parent transcends basic self-care needs to keeping their child safe in the wider world, often limiting the child's independence due to challenges with crossing roads which is a well-documented challenge related to DCD [19]. This lack of independence also limited play options with peers due to the additional scaffolding required to enable children with DCD to participate in physical activity in a meaningful and acceptable way, without judgement from their peers. Previous research has highlighted that children are 'onlookers' in the playground [28], but the current data revealed the hidden challenges these children faced with forming peer relationships due to their lack of ability to engage in age-appropriate play. Parents suggested this often impacted on the mental health and wellbeing of their child as they embodied a peripheral position within the social context. Parents alluded to low self-worth and confidence being a byproduct of their challenges associated with DCD, which often resulted in school-related anxiety. It is therefore unsurprising that previous research has linked DCD and poor sensorimotor development to worse mental health outcomes [15,29,30], and

an increased sense of victimisation [31]. This was evident in the current analysis when parents spoke about their child's experience of bullying. Interestingly though, parents revealed that their children formed strong, meaningful friendships with other children who had differences, often bonding due to shared challenges with peers.

Parents highlighted additional challenges associated with DCD, beyond the diagnostic criteria, related to cognition and learning, such as difficulties with memory and planning. Previous research has linked sensorimotor development to cognitive performance [32,18], however the current data revealed the extent to which these challenges impacted on children with DCD both in a school and home context. Parents revealed these underheard difficulties caused tension within and beyond the home, for example teachers not making reasonable adjustments, and wider family members becoming frustrated in relation to executive functioning challenges, despite strong parent advocates. A wider lack of knowledge and awareness was also seen as central to experiences, particularly in relation to teachers. This lack awareness in schools has been highlighted by previous research [9,33,34], however the current analysis illuminated the extent to which this lack of knowledge inhibited their child's needs being met. For example, for some the diagnosis was disregarded entirely, and even when this was not the case, reasonable adjustments were elusive when additional funding was not available. The strain on parents was evident when they had to fight for the rights of their children who were being discriminated against, which resulted in multiple school visits when, for example children were being victimised for eating too slowly in the dinner hall, or not being able to fasten their own coat, etc.

Despite strong efforts to advocate for their child, parents continued to meet resistance, during which time the support available in education for their child was sub-standard. In some cases, the struggle was such that parents felt that they had no choice but to remove their child from their current setting, either to home school, change schools or transition to a paid private school. Although some children were lucky enough to be born into families who had capacity to advocate, and could afford to pay for a private education, for many with low socioeconomic status backgrounds, this would not be possible [35]. In reality, children are beholden to a postcode lottery in terms of the support available both within healthcare and education [9,36].

As with previous research, the challenges did not stop with education – the healthcare systems both nationally and internationally were also seen as being highly barriered [9,10,37]. There were multiple accounts of frustration from parents who were beholden to a system which failed to provide adequate support. Parents alluded to a lack of knowledge amongst key healthcare professionals on their journey to receive a diagnosis, which sometimes resulted in multiple missed opportunities to identify their child's needs early. This affirms research suggesting a need for additional training to be embedded for all healthcare professionals involved in the DCD pathway [34]. Once their child received a diagnosis, parents illuminated that the challenges with healthcare continued, with limited resources, a lack of support beyond signposting, resulting in a sense of abandonment for families. The challenges with broken systems also resulted in some families having to find their own resources or pay for private assessment and intervention, often at great expense. Ultimately for families that could not afford to pursue this route, options were limited. The lack of freely available support is particularly concerning due to the potential for widening inequalities [38,39], which is compounded by the fact that early intervention is known to be effective for sensorimotor difficulties [8,40].

Parents highlighted their own lack of knowledge at the outset, with many lost in the unfamiliar DCD wilderness. Many relied on Google web searches to provide information, often with limited efficacy. This lack of knowledge was also alluded to being problematic within the wider community, due to common misconceptions of clumsiness. This could be lingering from previous diagnostic criteria [41], however recent research has shown that children were being given a clinical code of clumsiness as late as 2023 in Bradford, UK [11]. These misconceptions resulted in parents having to advocate for their child not only in healthcare and education, but also amongst wider family and friendship groups. When asked about wider community support (e.g., online and face to face groups), there was an overwhelming feeling that there was very little available support available, with some parents taking it upon themselves to collate or create resources for themselves and others. Participants also highlighted a stagnation of resources over time, with the lack of national charities for DCD in many countries leaving families without external support [42].

Despite the bleak story presented here, parents revealed that a diagnosis was central to building self-acceptance and confidence in their children. This aligns with previous literature which alludes to the importance of diagnosis for one's sense of self for neurodevelopmental disorders more broadly [43–45]. The current data presented the reality of the importance of having a label which allowed children to better understand themselves and articulate their struggles. Conversely, a label was not always considered favourable by the child, with reports of resistance to sharing through fear of potential stigmatisation, even amongst families.

### Limitations and future directions

There was no funding available to supply interpreters for this research, so only participants fluent in English were able to take part. This could mean that the data presented is reflective of a Westernised view of DCD, potentially skewing findings. In addition, as data collected was both contemporaneous (for families with recent diagnoses) and retrospective (for parents whose child was diagnosed many years ago). This could have led to differing accounts for a number of reasons, for example, accuracy of memory, and shifts in policy and practice [46]. Despite such variance, many of the parents interviewed experienced very similar barriers, which illuminates the stagnated nature of care for DCD. Whilst acknowledging the importance of the child voice in lived experience interviews [47,48], such research may be anxiety inducing given the sensitive nature of adverse experiences they have experienced. Furthermore, children are not always in a position to articulate effectively for themselves either contemporaneously or retrospectively [46].

Future research should endeavour to incorporate the voice of the child, and adults that have grown up with DCD to capture first-hand accounts of the challenges faced. In addition, future studies would benefit from taking a longitudinal approach to capture how the landscape changes across the lifespan. Finally, research would benefit from including lived experience from a wider range of countries, to better understand the global picture.

### Conclusion

Children with DCD and their families faced multiple profound challenges in relation to accessing help and support for their difficulties. It was evident that there is a need to raise the profile of DCD, but that limited knowledge means this should be facilitated via evidence-based training across sectors, which is underpinned by insider voices. In addition, DCD pathways need to be reviewed to ensure that children and their families feel adequately supported by education and healthcare contexts. For example, the community could play a greater role in supporting children with DCD and their families, however currently, families feel isolated and responsible for their own education (and at times the education of people managing their child's care), due to a lack of available resources. Fundamentally, more needs to be done to ensure all systems are working synergistically to allow all children with DCD the opportunity to thrive.

### Supporting information

**S1 File.  Agenda for parental interviews.**
(DOCX)

**S2 File.  COREQ (COnsolidated criteria for REporting Qualitative research) Checklist.**
(PDF)

### Author contributions

**Conceptualization:** Rebecca Murray, Cara E. Staniforth, Lucy H. Eddy.

**Data curation:** Rebecca Murray, Cara E. Staniforth, Amira Shawak, Lucy H. Eddy.

**Formal analysis:** Rebecca Murray, Cara E. Staniforth, Amira Shawak, Lucy H. Eddy.

**Funding acquisition:** Lucy H. Eddy.

**Investigation:** Rebecca Murray, Cara E. Staniforth, Amira Shawak, Lucy H. Eddy.

**Methodology:** Rebecca Murray, Cara E. Staniforth, Lucy H. Eddy.

**Project administration:** Rebecca Murray, Cara E. Staniforth, Lucy H. Eddy.

**Resources:** Rebecca Murray, Cara E. Staniforth, Lucy H. Eddy.

**Supervision:** Rebecca Murray, Cara E. Staniforth, Lucy H. Eddy.

**Writing – original draft:** Rebecca Murray.

**Writing – review & editing:** Rebecca Murray, Cara E. Staniforth, Amira Shawak, Lucy H. Eddy.

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
