## [Decision Letter · Decision Letter 0]

7 Aug 2025

PONE-D-25-28684Developmental Coordination Disorder: “It's not on people's radars… they're not interested”PLOS ONE

Dear Dr. Murray,

Thank you for submitting your manuscript to PLOS ONE. After careful consideration, we feel that it has merit but does not fully meet PLOS ONE’s publication criteria as it currently stands. Therefore, we invite you to submit a revised version of the manuscript that addresses the points raised during the review process. The reviewers have identified some areas of improvement, in particular in relation to reporting and framing of results, please have a look at these and address them appropriately. 

We look forward to receiving your revised manuscript.

Kind regards,

Aliah Faisal Shaheen

Academic Editor

PLOS ONE

Journal Requirements:

“The work of the senior author (L.H. Eddy) and co-author (C.E. Staniforth) was supported by a grant from the Waterloo Foundation (ref: 27665413).”

“The work of the senior author (L.H. Eddy) and co-author (C.E. Staniforth) was supported by a grant from the Waterloo Foundation (ref: 27665413).”

Reviewers' comments:

Reviewer's Responses to Questions

**Comments to the Author**

1. Is the manuscript technically sound, and do the data support the conclusions?

Reviewer #1: Yes

Reviewer #2: Yes

2. Has the statistical analysis been performed appropriately and rigorously?

Reviewer #1: N/A

Reviewer #2: N/A

3. Have the authors made all data underlying the findings in their manuscript fully available?

Reviewer #1: No

Reviewer #2: No

4. Is the manuscript presented in an intelligible fashion and written in standard English?

Reviewer #1: Yes

Reviewer #2: Yes

5. Review Comments to the Author

Reviewer #1: This is an interesting and insightful look into an overlooked condition. It is refreshing to see work going on in this area and I hope it spurs future work to improve the lives of these children and their families. However, there are a number of areas where the manuscript could be strengthened. Please find detailed comments below.

General

• Some minor grammar issues that can be remedied with a read through and check.

Abstract

• Just to highlight that there is quite a debate around the term ‘emergent’ or ‘emerging’ themes in qualitative research, particularly with regards to thematic analysis. Alternative descriptions could be: ‘identified’ or ‘developed’.

Introduction

• I think it would be good to give some concrete examples of the motor difficulties these children may experience to help give the reader a good understanding of exactly what DCD is.

• I feel you need more justification for why you have chosen to explore this in children, and particularly with parents – why not the children themselves?

Method

• I think it worth considering and discussing the possible impact on results that some of these parents will have been reporting current experiences versus other who will be looking back retrospectively.

• Identity-first language – i.e., autistic rather than with autism.

• How was the topic guide created? Was this based on previous literature or any existing models/theories/frameworks?

• Not sure if requested by this journal, but many suggest using the COREQ checklist for reporting qualitative research.

• “The authors formed the results section by drawing upon the most vibrant and compelling data, which was relevant to the research aims.” – Not quite sure what this means, please can you elaborate?

• “All authors were present when each transcript was analysed, which allowed consensus to be reached when disagreements and contrasting interpretations were tabled.” - Avoid 1 sentence paragraphs.

Results

• It would be good to start your results with a paragraph about the nature of your participants – you say they are culturally diverse, but there is no data showcasing this.

• The themes presented in your Abstract and Results section don’t exactly match up. Please keep consistent wording for clarity.

• Some formatting issues with the quotes in the results – please check and alter.

• It would be useful to have a table including your themes, sub-themes and examples quotes.

• “It appears that if the burden of diagnosis is to be alleviated for all children with DCD and they are to receive effective support across all sectors, including the family context. There is glaring a need for the DCD profile to be raised, which encompasses not just the diagnostic criteria but also the additional challenges faced by children with DCD and their families.” – I wonder if this is a Discussion point rather than results.

Discussion

• Remember to write using the past tense.

• Avoid informal language (e.g., ‘wasn’t’).

Reviewer #2: The manuscript “Developmental Coordination Disorder: “It's not on people's radars… they're not Interested” summarises a qualitative study to understand the experiences of children with DCD through parent-interviews. The work is well described and paper well written, with multifaceted by organised results that demonstrate the wide variety of difficulties experienced by children with DCD at a young age and their families. It particularly highlights and emphasises the need for more understanding of DCD, support for families in the medical and educational system, and the related challenges caused by misunderstanding or lack of knowledge surrounding DCD. I have only a few minor suggestions:

When discussing international differences in terminology, perhaps it is worth putting into the context that there was international consensus on using the term “DCD” since the 90s, yet this is not universally implemented.

Can some part of the introduction and/or limitations cover the use of parent-interviews in light of the “unheard voices” – why should this be more optimal than interviewing children directly?

Thank you for your interesting work!

6. PLOS authors have the option to publish the peer review history of their article (what does this mean?). If published, this will include your full peer review and any attached files.

Reviewer #1: No

Reviewer #2: No

---

## [Author Response · Author response to Decision Letter 1]

22 Aug 2025

Reviewer 1

This is an interesting and insightful look into an overlooked condition. It is refreshing to see work going on in this area and I hope it spurs future work to improve the lives of these children and their families. However, there are a number of areas where the manuscript could be strengthened. Please find detailed comments below.

We thank the reviewer for their kind words. We have worked hard to ensure the revised manuscript incorporates their helpful suggestions.

General

• Some minor grammar issues that can be remedied with a read through and check.

We have now been through the paper and resolved grammatical errors.

Abstract

• Just to highlight that there is quite a debate around the term ‘emergent’ or ‘emerging’ themes in qualitative research, particularly with regards to thematic analysis. Alternative descriptions could be: ‘identified’ or ‘developed’.

We thank the reviewer for flagging this contentious terminology – we have now changed this to ‘identified’ in the abstract and ensured that emergent and emerging do not appear throughout the manuscript.

Introduction

• I think it would be good to give some concrete examples of the motor difficulties these children may experience to help give the reader a good understanding of exactly what DCD is.

Thank you for highlighting this omission! We agree this is important for readers unfamiliar with DCD. We have now added a sentence in the introduction: These children often struggle with daily living skills such as dressing, personal hygiene, using cutlery, as well as engaging with physical activity and common education tasks such as handwriting [2,3].

• I feel you need more justification for why you have chosen to explore this in children, and particularly with parents – why not the children themselves?

We agree that there is a major need for children to be involved in lived experience. We have included a sentence in the introduction to justify our use of parent perspective as an alternative: Given the role that parents play as advocates for their child, it is crucial to access the parental perspective within the realm of the health and wellbeing of their child(ren).

We have also added more of a discussion around this in the limitations section to ensure full transparency: Whilst acknowledging the importance of the child voice in lived experience interviews [48,49], such research may be anxiety inducing given the sensitive nature of adverse experiences they have experienced. Furthermore, children are not always in a position to articulate effectively for themselves either contemporaneously or retrospectively [47].

Method

• I think it worth considering and discussing the possible impact on results that some of these parents will have been reporting current experiences versus other who will be looking back retrospectively.

We agree that it is certainly interesting to have both contemporary and retrospective experiences within this set of interviews. We have added a section on this to the limitations and future directions section of the discussion: ‘In addition, as data collected was both contemporaneous (for families with very recent diagnoses) and retrospective (for parents whose child was diagnosed many years ago), there is the potential for differing accounts for a number of reasons. For example, accuracy of memory, and shifts in policy and practice [47]. Despite such variance, many of the parents interviewed had experienced very similar barriers, which illuminates the stagnated nature of care for DCD.’

• Identity-first language – i.e., autistic rather than with autism.

We have now changed Autism to ‘Autistic children’ throughout.

• How was the topic guide created? Was this based on previous literature or any existing models/theories/frameworks?

We agree transparency around how the discussion guide was developed would strengthen the manuscript! We have added a section to the methods to incorporate this: Therefore the topic guide was intentionally broad in order to capture insights related to the challenges faced by children with DCD and their families according to previous literature [10,24], whilst also enabling a more holistic lens in order to explore underheard challenges.

• Not sure if requested by this journal, but many suggest using the COREQ checklist for reporting qualitative research.

After reviewing PLOS ONE guidelines, we have included the COREQ checklist for qualitative research as a supplementary file. We thank the reviewer for highlighting the need for a reporting checklist!

• “The authors formed the results section by drawing upon the most vibrant and compelling data, which was relevant to the research aims.” – Not quite sure what this means, please can you elaborate?

We have amended this text to clarify the meaning of the sentence within the manuscript: The quotes included in the results section were selected based on relevance to research aims, and their vibrant and compelling nature, which effectively articulated the lived experiences which transcended participants’ data.

• “All authors were present when each transcript was analysed, which allowed consensus to be reached when disagreements and contrasting interpretations were tabled.” - Avoid 1 sentence paragraphs.

We have now moved this sentence to join with the above paragraph within the manuscript.

Results

• It would be good to start your results with a paragraph about the nature of your participants – you say they are culturally diverse, but there is no data showcasing this.

We agree! We have a varied participant base, however due to small population with DCD and the personal stories shared, there is the potential for anonymity breaches. We have included additional locational information about these participants in the methods section, without being specific by country: Of these participants, two were from North America, one was from Asia and the remaining seven were based in the UK.

• The themes presented in your Abstract and Results section don’t exactly match up. Please keep consistent wording for clarity.

We thank the reviewer for highlighting this! We have changed the abstract to match the major themes presented in text: Identified major themes highlighted (i) impact of DCD on the lives of children and their families; (ii) lack of knowledge and awareness surrounding DCD; and (iii) inadequate support; and (iv) the role of diagnosis.

• Some formatting issues with the quotes in the results – please check and alter.

We have now ensured all quotes are formatted using the same style!

• It would be useful to have a table including your themes, sub-themes and examples quotes.

We have now included these (Table 1). We thank the reviewer as we believe this will help readers navigate a complex results section!

• “It appears that if the burden of diagnosis is to be alleviated for all children with DCD and they are to receive effective support across all sectors, including the family context. There is glaring a need for the DCD profile to be raised, which encompasses not just the diagnostic criteria but also the additional challenges faced by children with DCD and their families.” – I wonder if this is a Discussion point rather than results.

We agree this was going beyond the data and resembles more a discussion point. We have therefore removed this from results and incorporated in conclusion instead.

Discussion

• Remember to write using the past tense.

• Avoid informal language (e.g., ‘wasn’t’).

We have now amended the above errors in the manuscript!

Reviewer 2

The manuscript “Developmental Coordination Disorder: “It's not on people's radars… they're not Interested” summarises a qualitative study to understand the experiences of children with DCD through parent-interviews. The work is well described and paper well written, with multifaceted by organised results that demonstrate the wide variety of difficulties experienced by children with DCD at a young age and their families. It particularly highlights and emphasises the need for more understanding of DCD, support for families in the medical and educational system, and the related challenges caused by misunderstanding or lack of knowledge surrounding DCD. I have only a few minor suggestions:

We thank the reviewer for their words of support for this manuscript. We have revised the manuscript in line with the reviewer’s comments and are grateful for the improvements they have made to the paper!

When discussing international differences in terminology, perhaps it is worth putting into the context that there was international consensus on using the term “DCD” since the 90s, yet this is not universally implemented.

We agree this is an important point to make! We have added this to the introduction: International consensus was reached to use the term ‘DCD’ [8], however this is still inconsistently used.

Can some part of the introduction and/or limitations cover the use of parent-interviews in light of the “unheard voices” – why should this be more optimal than interviewing children directly?

We agree that there is a major need for children to be involved in lived experience. We have included a sentence in the introduction to justify our use of parent perspective as an alternative: Given the role that parents play as advocates for their child, it is crucial to access the parental perspective within the realm of the health and wellbeing of their child(ren).

We have also added more of a discussion around this in the limitations section to ensure full transparency: Whilst acknowledging the importance of the child voice in lived experience interviews [48,49], such research may be anxiety inducing given the sensitive nature of adverse experiences they have experienced. Furthermore, children are not always in a position to articulate effectively for themselves either contemporaneously or retrospectively [47].

Thank you for your interesting work!

We would like to thank the reviewer – your positivity is greatly appreciated!

---

## [Decision Letter · Decision Letter 1]

19 Sep 2025

PONE-D-25-28684R1Developmental Coordination Disorder: “It's not on people's radars… they're not interested”PLOS ONE

Dear Dr. Murray,

Thank you for submitting your manuscript to PLOS ONE. After careful consideration, we feel that it has merit but does not fully meet PLOS ONE’s publication criteria as it currently stands. Therefore, we invite you to submit a revised version of the manuscript that addresses the points raised during the review process.

The reviewers were generally supportive of the publication. One of the reviewers have pointed out a small improvement with relation to the subthemes - please review and resubmit. 

We look forward to receiving your revised manuscript.

Kind regards,

Aliah Faisal Shaheen

Academic Editor

PLOS ONE

Journal Requirements:

Reviewers' comments:

Reviewer's Responses to Questions

**Comments to the Author**

1. If the authors have adequately addressed your comments raised in a previous round of review and you feel that this manuscript is now acceptable for publication, you may indicate that here to bypass the “Comments to the Author” section, enter your conflict of interest statement in the “Confidential to Editor” section, and submit your "Accept" recommendation.

Reviewer #1: All comments have been addressed

Reviewer #2: All comments have been addressed

2. Is the manuscript technically sound, and do the data support the conclusions?

Reviewer #1: Yes

Reviewer #2: Yes

3. Has the statistical analysis been performed appropriately and rigorously?

Reviewer #1: N/A

Reviewer #2: Yes

4. Have the authors made all data underlying the findings in their manuscript fully available?

Reviewer #1: No

Reviewer #2: Yes

5. Is the manuscript presented in an intelligible fashion and written in standard English?

Reviewer #1: Yes

Reviewer #2: Yes

6. Review Comments to the Author

Reviewer #1: Thank you for addressing the reviewer's comments.

There is just one minor revision I would request. Currently, there are two sub-themes called 'Healthcare'. I would change one of these so they are distinctive.

Reviewer #2: The revision shows a number of improvements, sufficient consideration of reviewer concerns, and the new table helps improve the readability.

7. PLOS authors have the option to publish the peer review history of their article (what does this mean?). If published, this will include your full peer review and any attached files.

Reviewer #1: No

Reviewer #2: No

---

## [Author Response · Author response to Decision Letter 2]

23 Sep 2025

Reviewer 1

Thank you for addressing the reviewer's comments.

There is just one minor revision I would request. Currently, there are two sub-themes called 'Healthcare'. I would change one of these so they are distinctive.

We thank the reviewer for highlighting duplicated subtheme names. We have now amended these as follows:

Inadequate knowledge and awareness within Healthcare

Inadequate support within Healthcare

Reviewer 2

The revision shows a number of improvements, sufficient consideration of reviewer concerns, and the new table helps improve the readability.

We thank the reviewer for their kind words.

---

## [Editor Report · Decision Letter 2]

4 Nov 2025

Developmental Coordination Disorder: “It's not on people's radars… they're not interested”

PONE-D-25-28684R2

Dear Dr. Murray,

We’re pleased to inform you that your manuscript has been judged scientifically suitable for publication and will be formally accepted for publication once it meets all outstanding technical requirements.

Kind regards,

Aliah Faisal Shaheen

Academic Editor

PLOS ONE
---

## [Editor Report · Acceptance letter]

PONE-D-25-28684R2

PLOS ONE

Dear Dr. Murray,

I'm pleased to inform you that your manuscript has been deemed suitable for publication in PLOS ONE. Congratulations! Your manuscript is now being handed over to our production team.

Kind regards,

on behalf of

Dr. Aliah Faisal Shaheen

Academic Editor

PLOS ONE